# Microglial-stimulation of glioma invasion involves the EGFR ligand amphiregulin

Salvatore J. Coniglio[1,2]*, Jeffrey E. Segall[2,3]

**1** New Jersey Center for Science Technology and Mathematics, Kean University, Union, NJ, United States of America, **2** Department of Anatomy and Structural Biology, Albert Einstein College of Medicine, Bronx, NY, United States of America, **3** Gruss Lipper Biophotonics Center, Bronx, NY, United States of America

* coniglsa@kean.edu

**Data Availability Statement:** All relevant data are within the manuscript and its Supporting Information files.

**Funding:** The funders had no role in study design, data collection and analysis, decision to publish, or preparation of the manuscript.

## Abstract

High grade glioma is one of the deadliest human cancers with a median survival rate of only one year following diagnosis. The highly motile and invasive nature of high grade glioma makes it difficult to completely remove surgically. Therefore, increasing our knowledge of the mechanisms glioma cells use to invade normal brain is of critical importance in designing novel therapies. It was previously shown by our laboratory that tumor-associated microglia (TAMs) stimulate glioma cell invasion and this process is dependent on CSF-1R signaling. In this study, we seek to identify pro-invasive factors that are upregulated in microglia in a CSF-1R-dependent manner. We assayed cDNA and protein from microglia treated with conditioned media from the murine glioma cell line GL261, and discovered that several EGFR ligands including amphiregulin (AREG) are strongly upregulated. This upregulation is blocked by addition of a pharmacological CSF-1R inhibitor. Using RNA interference, we show that AREG-depleted microglia are less effective at promoting invasion of GL261 cells into Matrigel-coated invasion chambers. In addition, an AREG blocking antibody strongly attenuates the ability of THP-1 macrophages to activate human glioma cell line U87 invasion. Furthermore, we have identified a signaling pathway which involves CSF-1 signaling through ERK to upregulate AREG expression in microglia. Interfering with ERK using pharmacological inhibitors prevents AREG upregulation in microglia and microglia-stimulated GL261 invasion. These data highlight AREG as a key factor in produced by tumor associated microglia in promoting glioma invasion.

## Introduction

High grade glioma is an aggressive human cancer for which there is almost no effective treatment. One of the major characteristics of gliomais that it is highly motile and invasive. Glioma tumors have diffuse borders and are almost impossible to completely resect by surgery [1–3]. Therefore, understanding the mechanism of glioma invasion is of critical importance to ultimately discover more targeted therapy. Our laboratory and others have shown that microglia (macrophages that reside in the brain) are able to significantly enhance glioma cell invasion [4–16]. We previously published that microglial-stimulation of glioma invasion was almost

**Competing interests:** The authors have declared that no competing interests exist.

completely inhibited by pharmacological inhibition of the CSF-1 receptor (CSF-1R) [4]. That study showed that CSF-1 is expressed by GL261 glioma cells and the receptor CSF-1R is expressed by microglia, thus defining a paracrine interaction that takes place between glioma and microglia during invasion. It was found that by administering PLX-3397, a CSF-1R antagonist that can cross the blood brain barrier, the number of tumor-associated microglia was drastically reduced. In addition, tumors in animals treated with PLX-3397 exhibited substantially less invasion. For many sections of tumors in drug treated animals a clear border could be seen between the tumor and parenchyma.

EGFR is a receptor tyrosine kinase which is mutated or dysregulated in many cancers, especially glioma [17–21]. When active, EGFR transduces signals in the cell that stimulate proliferation, survival and motility [22–26]. The biology of EGFR is complex however, since in mammals, there are at least seven ligands that have been thus far identified which show the ability to bind and activate EGFR [27–30]. The situation is further complicated by the fact that the ligands themselves go through an extensive trafficking pattern in the cell which ultimately results in their release from the cell [31–34]. Ligands have been shown to function in an autocrine, paracrine and juxtacrine fashion [35–37]. Furthermore, evidence is accumulating that the ligands themselves can participate in "back" signaling [38–44]. Although each of the seven mammalian ligands use the EGFR, there is evidence for specific functions. For example Heparin Binding EGF (HB-EGF) is expressed in a subset of malignant gliomas and is required for tumor formation in a PTEN/INK4a-/- background mouse [45]. The ligand Betacellulin (BTC) was recently shown to drive glioma resistance to anti-STAT3 therapy [46].

The ligand amphiregulin (AREG) is one of seven ligands capable of binding and activating EGFR. AREG is synthesized as a 252 amino-acid precursor that is associated with the cell membrane. As with other EGFR ligands, AREG can be processed by proteases which results in release of the soluble "mature" ligand containing the EGF domain necessary for binding and triggering receptor dimerization. However, full length AREG associated with exosomes has been shown to promote invasion of breast carcinoma cells [47]. Interestingly, that study demonstrated that full length membrane-bound AREG deployed on exosomes stimulates breast carcinoma cell invasion to a greater extent than processed AREG or either form of other ligands. In addition to acting on EGFR expressed on tumor cells, AREG has recently been shown to promote differentiation of T cells into TREGs within the tumor microenvironment [48]. These observations suggest targeting AREG/EGFR in the tumor microenvironment may impact several compartments within the tumor microenvironment and could inhibit both tumor invasion and immunosuppression.

In the present study, we investigated the ability of glioma cell line conditioned media to stimulate expression of all known seven EGFR ligands in the mammalian genome. We focused our attention on the ligand Amphiregulin and its potential function in promoting glioma invasion.

## Materials and methods

### Cell culture and reagents

Murine microglia were derived from a spontaneously immortalized murine microglia cell cultures originally isolated from C57Bl/6J mice as previously described in Dobrenis et al. [49]. Briefly, primary microglia cultures were generated from high density mixed cell-type cultures of neonatal neocortex by differential adhesion methods producing highly purified (>99%) microglial populations as assessed by cell type specific markers including F4/80. To further maximize and ensure purity for experiments, the isolated cells were subcultured an additional 3 times with stringent selective adhesion on non-tissue culture-treated "suspension cell" plates

(Sarstedt) to further limit non-microglial cells. All cultures were maintained in Macrophage Serum-Free Medium (M-SFM; Invitrogen Cat# 12065–074) with 10% fetal calf serum. Microglia were supplemented with 10 ng/ml recombinant mouse granulocyte macrophage-colony stimulating factor (GM-CSF; R&D systems Cat #415-ML-10). All cells were cultured in a humidified incubator containing 5% $CO_2$ at 37 degrees. The cell lines used in this paper were GL261 murine glioma (obtained from NCI), U87 human glioma (ATCC), THP-1 human macrophage cell line (ATCC). Recombinant human CSF-1 was a gift from Chiron Corp. A CSF-1R receptor inhibitor, 4-Cyano-1H-pyrrole-2-carboxylic acid [4-(4-methyl-piperazin -1-yl)-2-(4-methyl-piperidin-1-yl)-phenyl]-amide provided by Johnson and Johnson Pharmaceutical Research and Development (referred to as JnJ [50, 51]) was used at 10 nM. Small interfering RNA (siRNA) si-GENOME duplexes targeting mouse amphiregulin were acquired from Dharmacon. Microglia were transfected using 2 ul of Dharmafect Reagent #1 with 20 nM siGENOME siRNA against murine amphiregulin (Dharmacon/Thermoscientific). Microglia were seeded 24 hours prior to transfection in a 6 well plate at 70% confluency. The siRNA and Dharmafect mixture was added to 1.6 mL of complete microglia growth media (MSFM with 10% FBS and 10ng/ml GMCSF) and added to the cells. Cells were incubated with transfection mix for 72 hours prior to experiment.

## Intracranial injection of glioma cells and isolation of microglia

All procedures involving mice were conducted in accordance with the National Institutes of Health regulations concerning the use and care of experimental animals. The study of mice was approved by the Albert Einstein College of Medicine animal use committee. For intracranial injection, C57BL/6J mice (10–12 weeks old; Jackson) were anesthetized with isofluorane. A hole was made 1 mm lateral and 2 mm anterior from the intersection of the coronal and sagittal sutures (bregma). 2 X 10 ^4 GL261 cells were injected using a Hamilton syringe series 7000 at a depth of approximately 1 mm in a volume of 0.2 μl in the cortex. Typically two weeks intracranial growth of GL261 did not result in any overt stress or apparent discomfort to the mice. Two weeks following injection, animals were anesthetized with isoflourane and sacrificed by cervical dislocation followed bytumor associated microglia isolation using Miltenyi Micro MACS system with CD11b microbeads (Catalog# 130-049-601 Miltenyi Biotec).

## Quantitative RT-PCR

Total RNA was isolated from cells in culture using according to manufacturer protocol of Mini RNeasy Kit (Qiagen). Total RNA was used as a template for cDNA synthesis prepared using Superscript II Reverse Transcriptase kit (Thermofisher). This material was subject to quantitative real time PCR using the following specific primer sets: EGF FWD: 5′-TTGT TAGCACCAT CCCTCAT-3′, REV: 5′-CGGGAGAGTTCTTTGTCTCA-3′, AREG FWD:5′AACGGTGTG GAGAAAAATCC3′ AREG REV:5′TTGTCCTCAGCTAGGCAATG3′ EREG FWD:5′TTTCTT CGTCCTTTGTTTGC-3′ EREG REV:5′CATATGCCAGGAAA AAGGTG-3′

TGFA FWD: 5′CACTGGACTTCAGCCCTCTA-3′ TGFA REV:5′-TCCAGCAGACCA GAAAAGAC-3;BTC FWD: 5′-GTGTGGTAGCAGATGGGAAC-3′,BTC REV: 5′ATCTCC CATGGATGCAGTAA3′-3′ GAPDH FWD: 5′-CTGGAGAAACCTGCCAAGTA-3′, GAPDH REV: 5′TGTTGCTGTAGCCGTATTCA-3′. Primers for HBEGF and EPGN were obtained from Qiagen (Mouse HBEGF:#PPM05369D, Mouse EPGN: PPM32906F). Reactions were carried out using SYBR Green PCR Master Mix (Qiagen) and the ABI SuperArray PA-012 (SABiosciences, Frederick, MD, USA).

## Invasion assays

Cell invasion assays were performed as previously described in [4, 52]. Briefly, GL261 and microglial (MG) cells were stained with cell tracker green (CMFDA) and with cell tracker red (CMTPX; Invitrogen) respectively and then cocultured on Matrigel-coated invasion chambers (Fisher Scientific #354480). For most assays, to maintain constant cell numbers, cells were plated at a density per invasion chamber of 100,000 labeled GL261 cells with an additional 100,000 unlabelled GL261 cells or with 50,000 unlabelled GL261 and 50,000 MG cells in M-SFM with 0.3% BSA (Sigma Aldrich #A9647). For measuring human glioma cell invasion, the human monocyte cell line THP-1 was differentiated with RPMI1 with 10% FBS and 100 nM phorbol 12-myristate 13-acetate (PMA, Sigma Aldrich Cat# P8139) for 48 hours and cultured in media alone for another 48 hours. U87 glioma cells were stained with green Cell Tracker Dye CMFDA (Invitrogen; Cat# C2925) and cocultured on invasion chambers as described above using 75,000 cells with 25,000 differentiated THP1 cells in RPMI with 0.3% BSA. Invasion chambers were incubated for 48 hours, after which they were fixed in 3.7% paraformaldehyde in PBS. Imaging of the cells on the bottom of the filter was performed using a Leica SP5 Laser Confocal System. The extent of invasion was quantified by counting the number of fluorescent glioma cells that were on the underside of the filter in at least seven 20X fields.

## Western blotting

Cell cultures of microglia starved overnight in M-SFM media (Invitrogen) 0.3% BSA and microglia stimulated with CSF-1 in the presence of various inhibitors were lysed directly into 1X sample buffer (2% SDS, 10% glycerol, 62.5 mM TRIS) containing beta mercaptoethanol (β-ME) and loaded onto 10% SDS PAGE gels. The proteins were resolved and transferred to a PVDF membrane and blotted using an anti-phospho CSF-1R Y723 (#3155; Cell Signaling Technology), or anti-phospho ERK antibody (#9101;Cell Signaling Technology), followed by secondary blot using anti-Rabbit conjugated to IR800 in Licor Blocking Buffer (Licor). For detecting EGF, blotting was carried out using anti-EGF (sc-1342; Santa Cruz) at a concentration of 1:100 followed by secondary anti-Goat conjugated to IR800 (Licor). For detecting CSF-1R, blotting was carried out using a rabbit antibody which recognizes the C-terminus of the CSF-1R (C-15;). Blots were scanned on the Odyssey system. Blots were stained with anti total ERK (#137F5; Cell Signaling Technology) and actin (A5441; Sigma-Aldrich) for loading control.

## Results

### Glioma induces AREG expression in microglia/macrophages

We have previously shown that microglia stimulate invasion of murine and human glioma cell lines and this is dependent on CSF-1R and EGFR signaling in a putative paracrine interaction [4]. We next wanted to measure the influence of glioma cells on the microglial expression of EGFR ligands. Our initial experimental approach was to treat microglia with conditioned media harvested from the murine glioma cell line, GL261 (referred to as GLCM). We then isolated RNA from microglia stimulated overnight and generated cDNA for quantitative PCR analysis. Primer sets specific for each of the seven EGFR ligands were used to measure the relative expression of EGF, HB-EGF, AREG (Amphiregulin), EREG (Epiregulin), TGF-A, BTC (Betacellulin) and EPN (Epigen). We detected a statistically significant induction in three of the seven EGFR ligands in microglia treated with GLCM: AREG, EREG and TGFA ([Fig 1A]). We measured the ability of glioma cells to stimulate AREG expression in microglia in coculture. Microglia were labelled with CMFDA green and cultured with GL261 cells for 24 hours followed by FACS sorting of microglia. When cocultured with GL261 cells, microglial expression

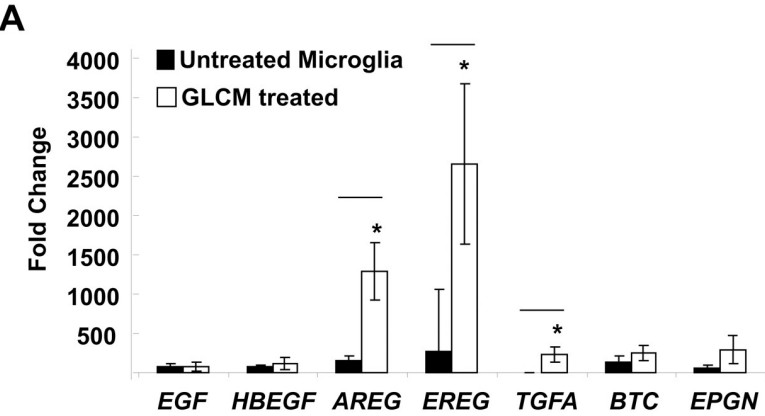

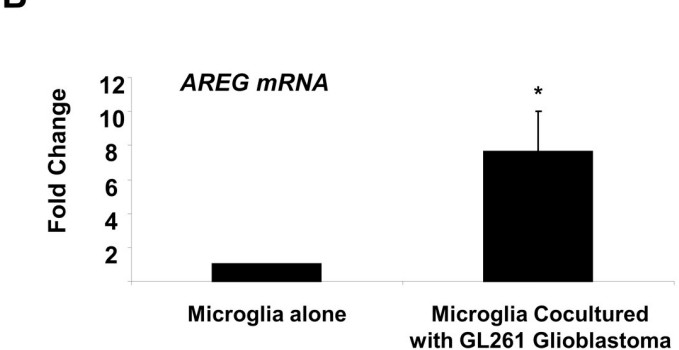

**Fig 1. EGFR ligand gene expression induced in microglia by glioma cells.** (A) Conditioned media harvested from GL261 cells was used to treat microglia overnight. Quantitative RT PCR was performed using the primers indicated. Data shown are $^2$ -delta delta ct relative to GDH control. *: P < .05. (B) GL261 cells were cocultured with microglia for 24 hours followed by cell sorting, RNA isolation and qRTPCR analysis using the AREG primers. Results are an average of 8 independent experiments *: P < 0.05.

of AREG is induced to a similar extent as what is observed with conditioned media (Fig 1B). We then assessed which of these ligands were induced in tumor associated microglia/macrophages (TAMs) in vivo. Two weeks after implantation, GL261 tumors were harvested from mice and the TAM population was isolated using magnetic sorting for CD11b positive cells. We harvested mRNA from CD11b positive (TAM) fraction and assessed the levels of AREG, EREG and TGFA as compared with freshly isolated naïve microglia from the brains of wild type C57 mice. Only AREG was induced in TAMs relative to wild type (naïve) microglia. In naïve microglia isolated from wild type C57BL/6 mice, AREG levels were undetectable by qPCR. However in CD11b+ cells isolated from GL261 tumors, we detected AREG with a mean dct value of 5.3 -/+ SEM of 3 in four independent experiments (four independent tumors). We were unable to detect EREG levels in naïve microglia and in TAMs. TGFa expression was not observed to increase in TAMs relative to naïve microglia. Having demonstrated that AREG expression is upregulated in TAMs in-vivo, we decided to focus our efforts on this EGFR ligand.

## AREG induction in microglia is dependent on CSF-1R

Since microglia stimulation of GL261 invasion is dependent on CSF-1R we reasoned that factors upregulated in microglia might be sensitive to CSF-1R inhibition. To test this, we treated

microglia with GLCM in the presence of a CSF-1R inhibitor (JnJ). Blockade of CSF-1R strongly attenuated (but did not fully inhibit) the ability of GLCM to stimulate AREG mRNA expression in microglia (Fig 2A). Interestingly, we found that recombinant CSF-1 alone could not stimulate AREG mRNA in microglia to levels seen with GLCM (Fig 2B). These data demonstrate that CSF-1 is a necessary factor generated by GL261 to induce AREG expression in microglia, but it is insufficient on its own. The level of AREG protein was ascertained using SDS-PAGE of microglial protein extracts followed by western blotting with an anti-AREG antibody (Fig 2C). The pattern of AREG protein expression was consistent what was seen at the mRNA level. The species of AREG we detect by western blot in microglia is almost exclusively the full-length and presumably the membrane associated form.

## AREG is involved in microglia/macrophage stimulation of glioma invasion

Next, we wished to test the functional significance of AREG induction in microglia. Our first approach was to use RNA interference mediated depletion of AREG in microglia using both siRNA pools and individual oligos. The siRNA pool was able to knockdown AREG levels to under 50% that of control (Fig 3A). Microglia which were depleted of AREG were largely unable to stimulate GL261 invasion (Fig 3B). This was observed using individual oligos as well to similar effect (S1 Fig)

We also examined the role of AREG in a human *in vitro* glioma model. Consistent with the mouse (GL261) model, the human macrophage cell line, THP-1 differentiated with phorbol myristate acetate (PMA) is able to stimulate the invasion of U87 human glioma line when cocultured on a Matrigel-coated transwell (Fig 4). As seen in the murine model, the CSF-1R inhibitor JnJ is able to strongly attenuate block THP1- stimulated U87 invasion (S2 Fig). We confirmed that THP-1 cells, but not U87 cells, express AREG (data not shown). In this model

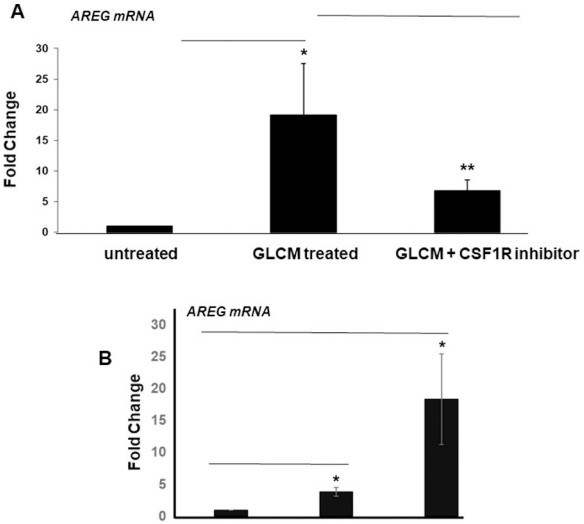

**Fig 2. Effect of CSF-1R inhibition on AREG induction in microglia.** (A) Conditioned media harvested from GL261 cells was used to treat microglia overnight in the absence or presence of 10 nM of CSF-1R inhibitor compound "JnJ" followed by RNA isolation, cDNA synthesis and qRTPCR analysis using the AREG primers. Results are an average of at least 3 independent experiments *: P < 0.05. (B) Microglia were treated with recombinant CSF-1 or GL261 conditioned media as described above followed by RNA isolation, cDNA synthesis and qRTPCR analysis using the AREG primers. Results are an average of at least 3 independent experiments *: P < 0.05. (C) SDS-PAGE analysis of AREG protein in microglia cell extracts treated with GL261 conditioned media in the absence or presence of 10 nM CSF-1R inhibitor (JnJ). Densitometry analysis carried out to evaluate level of AREG expression. Results are average of 3 independent experiments. *: P<0.05.

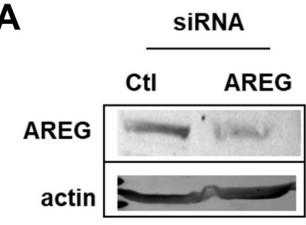

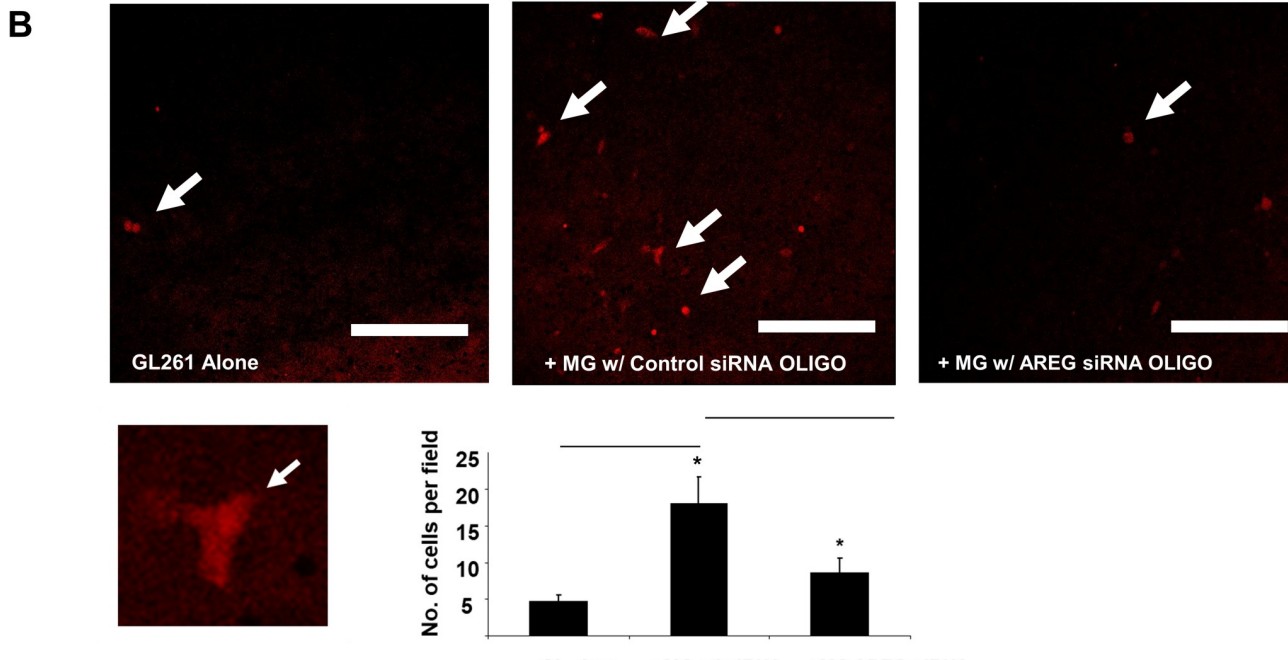

**Fig 3. Effect of AREG depletion in microglia-stimulated GL261 invasion.** (A) Murine AREG siRNA pool used to deplete AREG from microglial cells shows at least 50% knockdown at the protein level as determined by western blotting. (B) Microglial cells treated with either control or AREG siRNA pool were cocultured with GL261 cells expressing mCherry on Matrigel-coated invasion chambers. After 48 hours, transwells were fixed in formaldehyde and imaged using confocal microscopy. Representative images are shown. Bottom left micrograph is magnified image of an invasive mCherry-expressing GL261 tumor cell. Arrows indicate fluorescently labeled glioma cells which have invaded to the other side of the filter. Scale bar = 200 um. Results shown are average of at least three experiments. *: P < 0.05.

we used an alternative approach to interfere with AREG. An AREG function blocking antibody was included in the U87 + THP1 coculture and it was able to block THP1- stimulated U87 invasion to a very similar extent to that observed using RNAi against AREG in the murine GL261/microglial model. These data demonstrate that AREG is a key factor in microglia/macrophage promotion of glioma invasion in both mouse and human models.

## Glioma stimulation of AREG expression is dependent on the MAPK/ERK pathway

Thus far we have shown that the CSF-1/CSF-1R axis in microglia is required for AREG induction and stimulation of invasion. We next wanted to elucidate some of the signal transduction pathways that are involved in mediating CSF-1 activation of AREG transcription. Treatment of microglia with GLCM resulted in an increase in MAPK/ERK phosphorylation (Fig 5A). Interestingly, this was disrupted with addition of the CSF-1R inhibitor showing that CSF-1 is

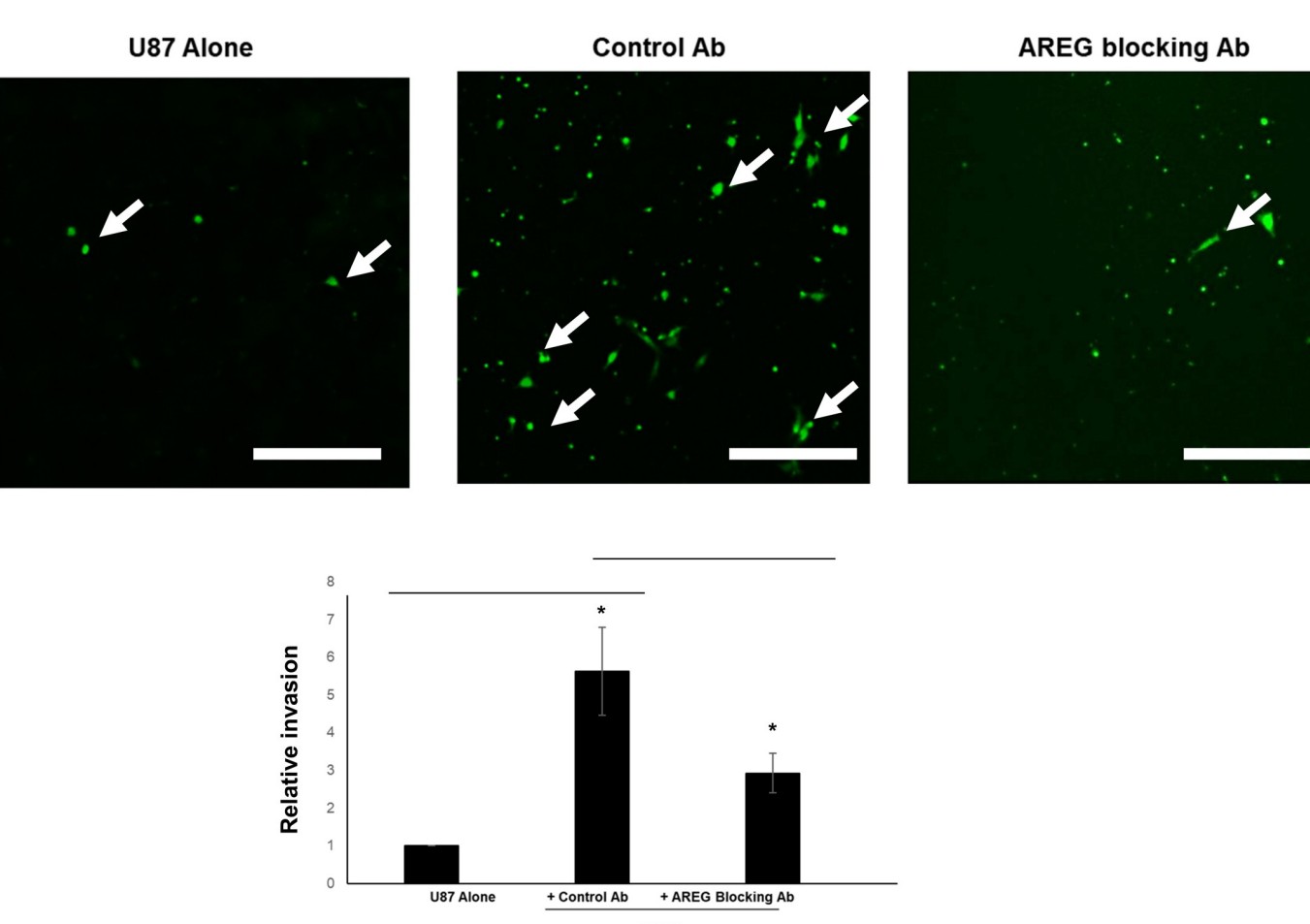

**Fig 4. Blockade of AREG from THP1 macrophages during U87 glioma invasion.** THP1 macrophages differentiated with PMA were cocultured with U87 cells stained with Cell Tracker Dye CMFDA (Green) on Matrigel-coated invasion with control IgG antibody or AREG blocking antibody. Representative images are shown. Arrows indicate fluorescently labeled glioma cells which have invaded to the other side of the filter. Scale bar = 200 um. Results shown are the average of at least three experiments. *: P < 0.05.

the predominant activator of ERK in microglia stimulated with GLCM (Fig 5A). We then wanted to assess the role of CSF-1/ERK signaling in GLCM stimulation of AREG transcription. Addition of U0126, an inhibitor of the MEK kinase which is upstream of ERK, was able to completely abolish GLCM induction of AREG mRNA (Fig 5B). We then tested the relevance of the ERK pathway in the coculture invasion assay. In the presence of the U0126 inhibitor, the ability of microglia to stimulate glioma invasion was strongly inhibited (Fig 5C). These data therefore show that CSF-1 via the ERK pathway induces AREG and the invasion promoting activity of microglial cells.

## Discussion

Our laboratory has demonstrated that the ability of glioma cells to invade is strongly increased when they are cocultured with microglia [4]. Furthermore, the ability of microglia to stimulate glioma invasion in this context is heavily dependent on CSF-1R signaling. This study addresses

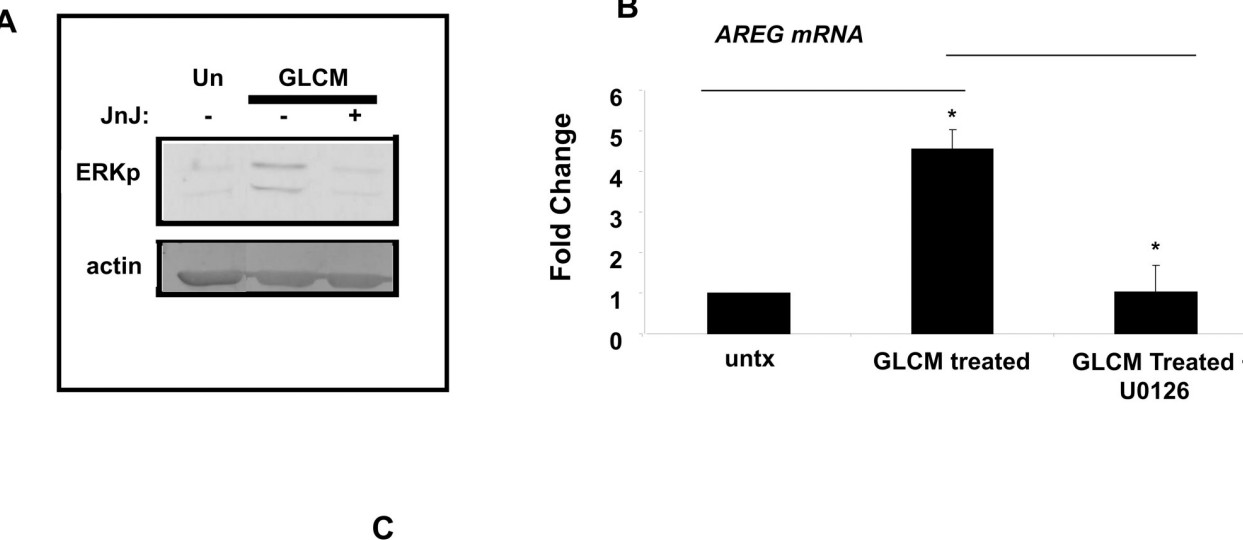

**Fig 5. Role of MEK/ERK in CSF1R dependent AREG induction and invasion.** (A) Microglia cells were stimulated with GL261 conditioned media in the absence or presence of the CSF1R inhibitor JnJ and analyzed by SDS-PAGE for phospho-ERK levels. (B) AREG expression in GL261-stimulated microglia was analyzed by qRTPCR in the presence of the MEK inhibitor U0126. (C) GL261 cells expressing mCherry were cocultured with microglia on Matrigel-coated invasion chambers in the absence or presence of the MEK inhibitor U0126. *:P<0.05.

the mechanism of the role of CSF-1 in this process. Here we show that conditioned media from GL261 cells strongly upregulates the EGFR ligand amphiregulin (AREG) in microglia in a CSF-1R dependent manner. This induction of AREG expression is important for cell invasion as interfering with AREG either by RNAi-mediated depletion or using function blocking antibodies, attenuates the ability of microglia and macrophages to stimulate glioma invasion.

This role of AREG in TAM-stimulated cell invasion is consistent with a previous study which assessed the ability of several EGFR ligands in promoting breast carcinoma cell invasion [47]. AREG has also been shown to function in a juxtacrine signaling [53]. Microglia apparently do not proteolytically process newly translated AREG as we could detect minimal AREG in the supernatant of stimulated microglia (data not shown). It is more likely that the precursor AREG we detect by western analysis remains associated with microglia by virtue of the fact that it contains a transmembrane domain. Consistent with this hypothesis, we could not stimulate glioma invasion to nearly the same extent using conditioned media from stimulated microglia (data not shown). This is also consistent what is observed in vivo where microglia are seen intimately connected to glioma cells at the invasive border.

The role of the CSF-1/CSF-1R axis in tumor progression is of great importance not just for glioma. Many other solid cancers have been shown to use the CSF-1 pathway to communicate

with TAMs during metastasis [54–58]. The "paracrine loop" interaction between breast carcinoma and tumor associated macrophages is well-documented [59–61]. In these models, secreted EGF is induced in a CSF-1 dependent manner and is released by macrophages [62]. The importance of CSF-1 in glioma been shown in a separate glioma model; where inhibition of CSF-1 results in the complete destruction of the tumor presumably by inducing the repolarization of TAMs from a trophic ("M2") state to a proinflammatory state ("M1") and thus reversing immunosuppression [63]. This was in an in-situ generated glioma mouse model using virus expressing PDGF to drive tumor formation. It is worth noting that this model replicates the "pro-neural" subtype of glioma which may not reflect other subtypes of glioma (such as the mesenchymal) [64–66]. It will be of critical importance to dissect the differences between these and other glioma subtypes with respect to how TAMs influence invasion and immunosuppression.

Signaling governed by CSF-1R in TAMs has been shown to promote immunosuppressive as well as invasive/metastatic behavior in malignant tumors. Consistent with AREG being a CSF-1R regulated gene, AREG also has pleiotropic roles during tumor development [67]. In addition to its canonical function in promoting EGFR activation on cancer cells, it also has the ability to influence the composition of the tumor microenvironment [48, 68, 69].

Here we show that expression of AREG is induced in TAMs by glioma in a CSF-1R dependent manner. The potential for using CSF-1R targeting drugs in combination with other therapies may hold great promise for treating glioma as well as other metastatic cancers [55, 70, 71].

## Supporting information

**S1 Fig. Effect of AREG depletion in microglia-stimulated GL261 invasion using individual AREG siRNA oligos.** Microglial cells depleted with either control or AREG siRNA individual oligos were cocultured with GL261 cells expressing mCherry on Matrigel-coated invasion chambers. Representative images are shown. Arrows indicate fluorescently labeled glioma cells which have invaded to the other side of the filter. Scale bar = 200 um. Results shown are average of at least five experiments. *: P < 0.05.
(TIF)

**S2 Fig. Effect of CSF-1R Iinhibition on THP-1 stimulation of U87 glioma THP1 macrophages differentiated with PMA were cocultured with U87 cells stained with cell tracker dye CMFDA (Green) on matrigel-coated invasion chambers in the absence or presence of 10 nM CSF1R inhibitor JnJ.** Results shown are average of at least three experiments. *: P < 0.05.
(TIF)

**S1 Dataset.**
(PDF)

**S1 Raw images.**
(PDF)

## Author Contributions

**Conceptualization:** Salvatore J. Coniglio.

**Data curation:** Jeffrey E. Segall.

**Formal analysis:** Salvatore J. Coniglio.

**Funding acquisition:** Salvatore J. Coniglio, Jeffrey E. Segall.

**Investigation:** Salvatore J. Coniglio.

**Methodology:** Salvatore J. Coniglio.

**Validation:** Salvatore J. Coniglio.

**Writing – original draft:** Salvatore J. Coniglio.

**Writing – review & editing:** Jeffrey E. Segall.

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
