## [Decision Letter · Decision Letter 0]

29 Jul 2021

PONE-D-21-18814

Microglial-stimulation of Glioblastoma Invasion Involves the EGFR ligand Amphiregulin

PLOS ONE

Dear Dr. Coniglio,

Thank you for submitting your manuscript to PLOS ONE. After careful consideration, we feel that it has merit but does not fully meet PLOS ONE’s publication criteria as it currently stands. Therefore, we invite you to submit a revised version of the manuscript that addresses the points raised during the review process.

We look forward to receiving your revised manuscript.

Kind regards,

Arun Rishi, Ph.D.

Academic Editor

PLOS ONE

2. To comply with PLOS ONE submissions requirements, in your Methods section, please provide additional information on the animal research and ensure you have included details on (1) methods of sacrifice, and (2) efforts to alleviate suffering.

3. To comply with PLOS ONE submissions requirements, in your Methods section, please provide additional information on the (1) Recombinant human CSF-1 and (2) CSF-1R receptor inhibitor, [4-Cyano-1H-pyrrole-2-carboxylic acid [4-(4-methyl-piperazin-1-yl)-2-(4-methyl-piperidin-1-yl)-phenyl]-amide].

When you resubmit, please ensure that you provide the correct grant numbers for the awards you received for your study in the ‘Funding Information’ section."

7. Please amend either the abstract on the online submission form (via Edit Submission) or the abstract in the manuscript so that they are identical.

8. PLOS ONE now requires that authors provide the original uncropped and unadjusted images underlying all blot or gel results reported in a submission’s figures or Supporting Information files. This policy and the journal’s other requirements for blot/gel reporting and figure preparation are described in detail at https://journals.plos.org/plosone/s/figures#loc-blot-and-gel-reporting-requirements and https://journals.plos.org/plosone/s/figures#loc-preparing-figures-from-image-files. When you submit your revised manuscript, please ensure that your figures adhere fully to these guidelines and provide the original underlying images for all blot or gel data reported in your submission. See the following link for instructions on providing the original image data: https://journals.plos.org/plosone/s/figures#loc-original-images-for-blots-and-gels

9 Please review your reference list to ensure that it is complete and correct. If you have cited papers that have been retracted, please include the rationale for doing so in the manuscript text, or remove these references and replace them with relevant current references. Any changes to the reference list should be mentioned in the rebuttal letter that accompanies your revised manuscript. If you need to cite a retracted article, indicate the article’s retracted status in the References list and also include a citation and full reference for the retraction notice.

Reviewers' comments:

Reviewer's Responses to Questions

**Comments to the Author**

1. Is the manuscript technically sound, and do the data support the conclusions?

Reviewer #1: Yes

Reviewer #2: Yes

2. Has the statistical analysis been performed appropriately and rigorously? 

Reviewer #1: Yes

Reviewer #2: Yes

3. Have the authors made all data underlying the findings in their manuscript fully available?

Reviewer #1: Yes

Reviewer #2: Yes

4. Is the manuscript presented in an intelligible fashion and written in standard English?

Reviewer #1: Yes

Reviewer #2: Yes

5. Review Comments to the Author

Reviewer #1: Overall opinion and general observations of the manuscript PONE-D-21-18814

Summary

This is a comprehensive study regarding the mechanisms glioblastoma (GBM) cells use to invade normal brain. It is known that microglia stimulate GBM cell invasion, and that this process is dependent on CSF-1R signaling. However, to have discovered more in-depth mechanism and pro-invasive factors upregulated in microglia, particularly amphiregulin (AREG), is interesting since targeting AREG/EGFR in the tumor could inhibit tumor invasion. Moreover, these extensive results are promising in terms of targeted therapies for GBM.

However, there are some concerns with regard to specific statements.

1. Term "multiforme" is no longer in use. According to the WHO 2016 classification of diffuse gliomas this term is omitted. Moreover, in Abstract you used this term with glioma, which is inappropriate.

2. You have cited plenty of references, however, only less than 20 is issued since 2015. For example, in Introduction you cited 8 references that showed that microglia are enhancers of glioma cell invasion, and none of them is issued since 2015.

3. The last paragraph in Introduction should be the aim of your research, not to conclude what you have found. Otherwise, it looks like Conclusion. This paragraph needs to be changed.

4. I wondered about the concentration of CSF-1R inhibitor (JnJ). Why did you use 10 nM?

5. Can you elaborate the place of inoculation of GBM cells? Why did you use those coordinates for induction of glioblastoma? Does it have impact on your model of glioblastoma?

6. Please, indicate magnification you used in Fig.3B and Fig.4. Also, if possible, add scale bar in both microphotographs. Explain what arrows indicate.

7. In Fig.4 the number of invaded cells in group treated with AREG blocking Ab is higher than in U87 alone, while in microphotographs it looks comparable between these two groups. Please, use consistent pictures.

Reviewer #2: This is an interesting article that builds on groups' prior published work from 2012, showing that CSF1-R inhibition in macrophage/microglia results in decreased glioma invasion. In the current manuscript, the authors show the mechanism by which CSF-1R signaling in microglia results in upregulation of AREG, leading to glioma cell invasion. They further show that interfering with AREG either by using RNAi or blocking antibodies attenuate macrophages/microglia's ability to promote glioma invasion. Overall, it is a well-controlled and concise in vitro study. There are a few issues that need to be fixed before considering it for publishing.

Figure 1B is missing the error bar for the microglia alone group, or if it is normalized per microglia alone group, in which case it should be fold change on Y-axis. This goes for Figure 2 as well.

Figure 2 in the text states four independent experiments – are those from four independent tumors? Need to be clarified.

In the text, it states, "Interestingly, we found that recombinant CSF-1 alone was insufficient to stimulate AREG mRNA in microglia to levels seen with GLCM (Fig 2B). Unfortunately, Figure 2B does not support the statement, and multiple comparison test should be used to see whether induction of AREG by CSF-1 stimulation and GLCM is in Figure 2B.

Images for invasion in all the figures are not very impressive, and some high-resolution inserts would be helpful.

Glioma Multiforme term is no longer used needs to be just glioblastoma.

6. PLOS authors have the option to publish the peer review history of their article (what does this mean?). If published, this will include your full peer review and any attached files.

Reviewer #1: No

Reviewer #2: No

---

## [Author Response · Author response to Decision Letter 0]

28 Oct 2021

Response to Reviewers

Editor Comments:

2. To comply with PLOS ONE submissions requirements, in your Methods section, please provide additional information on the animal research and ensure you have included details on (1) methods of sacrifice, and (2) efforts to alleviate suffering.

This description has been added to the materials and methods section

3. To comply with PLOS ONE submissions requirements, in your Methods section, please provide additional information on the (1) Recombinant human CSF-1 and (2) CSF-1R receptor inhibitor, [4-Cyano-1H-pyrrole-2-carboxylic acid [4-(4-methyl-piperazin-1-yl)-2-(4-methyl-piperidin-1-yl)-phenyl]-amide].

We have included the appropriate references for this compound in the revised manuscript.

4. We note that the grant information you provided in the ‘Funding Information’ and ‘Financial Disclosure’ sections do not match. When you resubmit, please ensure that you provide the correct grant numbers for the awards you received for your study in the ‘Funding Information’ section."

This information is now identical

That is fine.

ORCID ID was added to “My Information” page on the Manager site.

7. Please amend either the abstract on the online submission form (via Edit Submission) or the abstract in the manuscript so that they are identical.

The abstract in the online submission form and the manuscript are now identical

8. PLOS ONE now requires that authors provide the original uncropped and unadjusted images underlying all blot or gel results reported in a submission’s figures or Supporting Information files. This policy and the journal’s other requirements for blot/gel reporting and figure preparation are described in detail at https://journals.plos.org/plosone/s/figures#loc-blot-and-gel-reporting-requirements and https://journals.plos.org/plosone/s/figures#loc-preparing-figures-from-image-files. When you submit your revised manuscript, please ensure that your figures adhere fully to these guidelines and provide the original underlying images for all blot or gel data reported in your submission. See the following link for instructions on providing the original image data: https://journals.plos.org/plosone/s/figures#loc-original-images-for-blots-and-gels

The original blot images (uncropped) are included in the revised manuscript under supporting material in the file “S1_Raw-Images”. 

9 Please review your reference list to ensure that it is complete and correct. If you have cited papers that have been retracted, please include the rationale for doing so in the manuscript text, or remove these references and replace them with relevant current references. Any changes to the reference list should be mentioned in the rebuttal letter that accompanies your revised manuscript. If you need to cite a retracted article, indicate the article’s retracted status in the References list and also include a citation and full reference for the retraction notice.

As per reviewer request, several additional references have been added to the revised manuscript. The additional references are listed below:

Roos A, Ding Z, Loftus JC, Tran NL. Molecular and microenvironmental determinants of glioma stem-like cell survival and invasion. Frontiers in Oncology. 2017;7: 1–8. doi:10.3389/fonc.2017.00120

Masui K, Kato Y, Sawada T, Mischel PS, Shibata N. Molecular and genetic determinants of glioma cell invasion. International Journal of Molecular Sciences. 2017;18. doi:10.3390/ijms18122609

Wallmann T, Zhang XM, Wallerius M, Bolin S, Joly AL, Sobocki C, et al. Microglia Induce PDGFRB Expression in Glioma Cells to Enhance Their Migratory Capacity. iScience. 2018;9: 71–83. doi:10.1016/j.isci.2018.10.011

Zhang X, Chen L, Dang W qi, Cao M fu, Xiao J fang, Lv S qing, et al. CCL8 secreted by tumor-associated macrophages promotes invasion and stemness of glioblastoma cells via ERK1/2 signaling. Laboratory Investigation. 2019;8. doi:10.1038/s41374-019-0345-3

Manini I, Caponnetto F, Bartolini A, Ius T, Mariuzzi L, Loreto C di, et al. Role of microenvironment in glioma invasion: What we learned from in vitro models. International Journal of Molecular Sciences. 2018;19. doi:10.3390/ijms19010147

Illig CR, Chen J, Wall MJ, Wilson KJ, Ballentine SK, Rudolph MJ, et al. Discovery of novel FMS kinase inhibitors as anti-inflammatory agents. Bioorganic & medicinal chemistry letters. 2008;18: 1642–8. doi:10.1016/j.bmcl.2008.01.059

Manthey CL, Johnson DL, Illig CR, Tuman RW, Zhou Z, Baker JF, et al. JNJ-28312141, a novel orally active colony-stimulating factor-1 receptor/FMS-related receptor tyrosine kinase-3 receptor tyrosine kinase inhibitor with potential utility in solid tumors, bone metastases, and acute myeloid leukemia. Molecular cancer therapeutics. 2009;8: 3151–3161. doi:10.1158/1535-7163.MCT-09-0255

Reviewer #1 Comments:

1. Term "multiforme" is no longer in use. According to the WHO 2016 classification of diffuse gliomas this term is omitted. Moreover, in Abstract you used this term with glioma, which is inappropriate.

We have eliminated the term “multiforme” in the manuscript. All mention of such has been replaced with “high grade glioma” or simply “glioma”.

2. You have cited plenty of references, however, only less than 20 is issued since 2015. For example, in Introduction you cited 8 references that showed that microglia are enhancers of glioma cell invasion, and none of them is issued since 2015.

We have updated our references and included more recent studies demonstrating a role for microglia and tumor associated macrophages playing a role in glioma invasion and progression.

3. The last paragraph in Introduction should be the aim of your research, not to conclude what you have found. Otherwise, it looks like Conclusion. This paragraph needs to be changed.

We have altered the last paragraph of the introduction in the revised manuscript.

4. I wondered about the concentration of CSF-1R inhibitor (JnJ). Why did you use 10 nM?

This CSF-1R antagonist is quite potent as we discovered it can fully block CSF-1 stimulation of CSF-1R tyrosine phosphorylation (detected using western blot) at concentrations as low as 10 nM. This was published in our 2012 paper (See figure 3, Coniglio et al 2012 Mol Med (2012) 18:519–27. 

5. Can you elaborate the place of inoculation of GBM cells? Why did you use those coordinates for induction of glioblastoma? Does it have impact on your model of glioblastoma?

The location of tumor implantation was performed consistent with our previous publication. During our initial studies, we did not notice a significant dependence on tumor location with respect to macrophage infiltration and invasion.

6. Please, indicate magnification you used in Fig.3B and Fig.4. Also, if possible, add scale bar in both microphotographs. Explain what arrows indicate.

We have added scale bars to all of the images. The revised figure legend also now includes a statement that the arrows indicate invasive cells.

7. In Fig.4 the number of invaded cells in group treated with AREG blocking Ab is higher than in U87 alone, while in microphotographs it looks comparable between these two groups. Please, use consistent pictures.

We enhanced the image to show the invading cells in this condition. It now matches the quantitation well.

Reviewer #2 Comments:

1.Figure 1B is missing the error bar for the microglia alone group, or if it is normalized per microglia alone group, in which case it should be fold change on Y-axis. This goes for Figure 2 as well.

Yes, the qrtpcr results were normalized to unstimulated microglia. We therefore labelled the Y-axis label to indicate “Fold Change”

2.Figure 2 in the text states four independent experiments – are those from four independent tumors? Need to be clarified.

Yes this refers to four independent tumors. We have clarified this in the text.

3.In the text, it states, "Interestingly, we found that recombinant CSF-1 alone was insufficient to stimulate AREG mRNA in microglia to levels seen with GLCM (Fig 2B). Unfortunately, Figure 2B does not support the statement, and multiple comparison test should be used to see whether induction of AREG by CSF-1 stimulation and GLCM is in Figure 2B.

We realize the description of the data is confusing and we have changed the wording. Our assertion is that CSF-1 alone cannot achieve the level of AREG mRNA induction to levels observed using conditioned media (GLCM) The actual values are normalized to unstimulated microglia. Recombinant CSF-1 stimulation alone results in a 3.86 fold increase in AREG mRNA (standard error of 0.83) while using conditioned media from GL261 cells (GLCM) results in a 25.07 fold increase in AREG mRNA (standard error of 11.7). These data were obtained from an average of six independent experiments. Figure 2A shows that blockade of CSF-1R had a strong effect on GLCM stimulation of AREG mRNA. We conclude that CSF-1 is synergizing with an as of yet unindentified factor secreted by GL261 cells to promote full AREG mRNA induction in microglia.

4.Images for invasion in all the figures are not very impressive, and some high-resolution inserts would be helpful.

The images were all taken with a lower magnification to enable us count as many cells in as wide a field as possible. Included in the revised manuscript figures is a magnification of the area which shows the invasive cells in more detail.

5.Glioma Multiforme term is no longer used needs to be just glioblastoma.

We have removed the term “multiforme” throughout the text and generally use “glioma” or “high grade glioma”.

---

## [Editor Report · Decision Letter 1]

8 Nov 2021

Microglial Stimulation of Glioma Invasion Involves the EGFR ligand Amphiregulin

PONE-D-21-18814R1

Dear Dr. Coniglio,

We’re pleased to inform you that your manuscript has been judged scientifically suitable for publication and will be formally accepted for publication once it meets all outstanding technical requirements.

Kind regards,

Arun Rishi, Ph.D.

Academic Editor

PLOS ONE
---

## [Editor Report · Acceptance letter]

15 Nov 2021

PONE-D-21-18814R1 

Microglial-stimulation of Glioma Invasion Involves the EGFR ligand Amphiregulin 

Dear Dr. Coniglio:

I'm pleased to inform you that your manuscript has been deemed suitable for publication in PLOS ONE. Congratulations! Your manuscript is now with our production department. 

Kind regards, 

on behalf of

Prof Arun Rishi 

Academic Editor

PLOS ONE